# Recovering What Matters: High Protein Recovery after Endotoxin Removal from LPS-Contaminated Formulations Using Novel Anti-Lipid A Antibody Microparticle Conjugates

**DOI:** 10.3390/ijms241813971

**Published:** 2023-09-12

**Authors:** Cristiane Casonato Melo, Alexandra C. Fux, Martin Himly, Neus G. Bastús, Laura Schlahsa, Christiane Siewert, Victor Puntes, Albert Duschl, Isabel Gessner, Jonathan A. Fauerbach

**Affiliations:** 1Chemical Biology Department, R&D Reagents, Miltenyi Biotec B.V. & Co. KG, 51429 Bergisch Gladbach, Germany; cristianec@miltenyi.com (C.C.M.); alexandraf@miltenyi.com (A.C.F.); laurasch@miltenyi.com (L.S.); christianes@miltenyi.com (C.S.); 2Division of Allergy & Immunology, Department of Biosciences & Medical Biology, Paris Lodron University of Salzburg, 5020 Salzburg, Austria; martin.himly@plus.ac.at (M.H.); albert.duschl@plus.ac.at (A.D.); 3Institut Català de Nanociència i Nanotecnologia (ICN2), Consejo Superior de Investigaciones Científicas (CSIC), The Barcelona Institute of Science and Technology (BIST), Campus UAB, Bellaterra, 08036 Barcelona, Spain; neus.bastus@icn2.cat (N.G.B.); victor.puntes@icn2.cat (V.P.); 4Networking Research Centre on Bioengineering, Biomaterials and Nanomedicine (CIBER-BBN), 08034 Barcelona, Spain; 5Vall d’Hebron Institut de Recerca (VHIR), 08035 Barcelona, Spain; 6Institució Catalana de Recerca i Estudis Avançats (ICREA), 08010 Barcelona, Spain

**Keywords:** LPS contamination, polystyrene particles, bioconjugation, supramolecular structures, LAL, NTA

## Abstract

Endotoxins or lipopolysaccharides (LPS), found in the outer membrane of Gram-negative bacterial cell walls, can stimulate the human innate immune system, leading to life-threatening symptoms. Therefore, regulatory limits for endotoxin content apply to injectable pharmaceuticals, and excess LPS must be removed before commercialization. The majority of available endotoxin removal systems are based on the non-specific adsorption of LPS to charged and/or hydrophobic surfaces. Albeit effective to remove endotoxins, the lack of specificity can result in the unwanted loss of essential proteins from the pharmaceutical formulation. In this work, we developed microparticles conjugated to anti-Lipid A antibodies for selective endotoxin removal. Anti-Lipid A particles were characterized using flow cytometry and microscopy techniques. These particles exhibited a depletion capacity > 6 ×10^3^ endotoxin units/mg particles from water, as determined with two independent methods (Limulus Amebocyte Lysate test and nanoparticle tracking analysis). Additionally, we compared these particles with a non-specific endotoxin removal system in a series of formulations of increasing complexity: bovine serum albumin in water < insulin in buffer < birch pollen extracts. We demonstrated that the specific anti-Lipid A particles show a higher protein recovery without compromising their endotoxin removal capacity. Consequently, we believe that the specificity layer integrated by the anti-Lipid A antibody could be advantageous to enhance product yield.

## 1. Introduction

To survive in a diversity of hostile environments, bacteria have developed a complex cell envelope, with selective passage of substances [1]. In Gram-negative bacteria, a three-layer envelope (outer membrane, peptidoglycan cell wall and inner membrane) protects the inner content [1]. The outer membrane is mainly formed of lipopolysaccharides (LPS), which confer structural integrity. Moreover, it limits the translocation of hydrophobic molecules, such as during Gram staining [1,2]. LPS molecules are normally composed of two different chemical components responsible for their amphiphilic nature, the hydrophilic polysaccharide moiety and the hydrophobic Lipid A domain [2,3]. The latter is commonly considered the most conserved region of LPS among different bacterial species and strains and is known to be responsible for the activation of the human innate immune system when LPS enters the human bloodstream [2,3,4]. The intensity of the immune response against LPS can be different depending on the LPS source and concentration, but, in general, LPS has been described to trigger the release of inflammatory cytokines upon activation of the TLR4 pathway, with symptoms ranging from fever up to life-threatening conditions, such as septic shock or even death [5].

To avoid contamination incidents upon drug administration into patients, all injectable pharmaceuticals need to be tested for endotoxin content before being released for commercialization. With the exception of intrathecal administration, a limit of five endotoxin units (EU)/kg body weight per hour has been established for parenteral formulations [6,7]. Whenever the limit is exceeded, endotoxin inactivation or removal is required prior to safe release of the respective pharmaceutical product [8].

Nevertheless, the removal of LPS cannot be achieved through conventional sterilization methods such as 0.22 µm filtration and/or autoclaving [9,10]. Instead, the elimination of pyrogens, including endotoxins, a process also called depyrogenation, can be accomplished through specific methods such as using oxidizing agents (e.g., hydrogen peroxide), dry heat at 250 °C for at least 30 min or ionizing radiation [9,11]. However, such harsh conditions, although effective to eliminate endotoxins, are not compatible with many drug products. The need for alternative endotoxin removal methods has become even more evident as several biopharmaceutical proteins are produced recombinantly in *E. coli*, a Gram-negative bacterium, and endotoxin removal is therefore an integral part of their purification processes [6,12]. In these scenarios, the downstream purification process of such proteins can account for up to 92% of manufacturing costs [6]. 

To date, most of the commercially available endotoxin removal methods are based on non-specific interactions between positively charged surfaces and the negatively charged phosphate group in the LPS backbone and/or between hydrophobic surfaces and the hydrophobic Lipid A [6,13,14]. While these methods are effective in removing LPS from various aqueous solutions, their lack of specificity for LPS leads to undesirable co-removal of other molecules, such as proteins, resulting in low recovery and higher production costs [6]. To reduce the non-specific depletion of active components (e.g., proteins), alternative systems have been developed that rely on high-affinity interactions between LPS and a modified surface. For instance, Vagenende et al. described a system that relies on hydrogen bonding between endotoxins and crystals made of purine-derived allantoin that reaches binding affinities in the picomolar range. In addition to a good efficiency in LPS removal, a protein recovery of over 80% for bovine serum albumin (BSA) was achieved [15]. Li et al. decorated nanoparticles with lipopolysaccharide-binding proteins (LBP) and achieved a good LPS removal efficiency under controlled contamination conditions, with a stable protein recovery of 80% for BSA at pH values ranging from four to nine [16]. Recently, Chen et al. presented the development of silica microparticles conjugated to histidine dipeptides and showed its applicability to remove LPS from buffer, protein solution and from whole blood [17]. Similar results were shown by Shi et al. using artificial peptides and a hemocompatible polymer. Furthermore, they demonstrated LPS clearance in extracorporeal in vivo studies [18]. Despite these improvements in targeting LPS, we rationalized that, by means of an antibody–antigen interaction targeting the Lipid A in LPS, we could further optimize the specific removal of LPS while achieving a high protein yield.

In this work, we therefore present a new endotoxin removal platform based on the specific interaction between the Lipid A moiety from LPS and an anti-Lipid A antibody immobilized on the surface of polystyrene microparticles. We determine LPS removal in water using the well-established and widely used Limulus Amebocyte Lysate (LAL) test and compared the results to an alternatively developed method based on nanoparticle tracking analysis (NTA), which tracks the presence of supramolecular structures of LPS. While NTA is commonly used for the characterization of liposomes, lipidic nanoparticles and extracellular vesicles [19,20,21], it has, to the best of our knowledge, not yet been described for the evaluation of LPS supramolecular structures, with a clear advantage over LAL regarding sample preparation time. Additionally, we assessed the performance of anti-Lipid A particles versus a non-specific approach regarding LPS removal efficiency and protein recovery for a series of samples with increasing complexity: BSA in water, insulin in HEPES buffer and naturally LPS-contaminated birch pollen extracts (BPEs) in DPBS. We demonstrate that while both particle systems maintain a similar LPS removal performance, only the specific anti-Lipid A particles enabled a statistically significant higher protein recovery, consequently providing a safety and potential commercial advantage for pharmaceutical formulations. 

## 2. Results and Discussion

The objective of this study was to investigate whether the unwanted loss of proteins can be reduced by using LPS removal systems that specifically interact with LPS. We therefore designed a particle platform that was based on micron-sized polystyrene (PS) particles that were surface-modified with anti-Lipid A antibodies. 

### 2.1. Anti-Lipid A Particles: Conjugation and Characterization

Anti-Lipid A particles were prepared through a three-step conjugation as depicted in Figure 1a. In short, carboxylic acid groups on the surface of 3 µm PS particles were activated and covalently bound to aminodextran to yield amino-PS particles. Subsequent activation of primary amines with maleimides allowed for the attachment of reduced anti-Lipid A antibodies (Ab) via Michael addition.

After conjugation, anti-Lipid A particles were imaged using scanning transmission electron microscopy (STEM) and scanning electron microscopy (SEM) (Figure 1b and Appendix A), which revealed spherical particles with a narrow size distribution (2.68 ± 0.05 µm, Appendix A). Optical images obtained using Countess II Cell counter indicated a high dispersibility of the conjugated particles in phosphate-buffered saline (PBS)-0.03% Pluronic F-68 with no visual signs of aggregation (Appendix A). This was further substantiated by a high percentage value of single particles obtained using flow cytometry measurements (Appendix A). The supernatant (SN) obtained after Ab conjugation with maleimide particles was subjected to protein quantification via a bicinchoninic acid (BCA) assay, indicating that an average of 5.1 ± 0.6 µg (n = 3) of polyclonal anti-Lipid A Ab was conjugated to 1 × 10^8^ particles (Figure 1c). Upon the incubation of anti-Lipid A particles with a specific secondary Ab (anti-Goat-PE), it was possible to stain the conjugated Ab on the surface of the particles. An increase in the PE median fluorescence intensity (MFI) compared with controls indicated the presence of anti-Lipid A Ab on the particle surface, which provides an independent determination of Ab conjugation to that of the BCA assay. No fluorescence was observed for Ab-conjugated particles without secondary Ab (negative control) nor for Ab-conjugated particles incubated with an unspecific secondary Ab (anti-Human-PE, Figure 1d and Appendix A). 

### 2.2. Anti-Lipid A Particles’ Functionality and Stability 

#### 2.2.1. LPS Removal Capacity of Anti-Lipid A Particles 

Next, the LPS removal capacity of anti-Lipid A particles was determined. In view of the ubiquity of *E. coli* [22], a solution of 1000 ng/mL ≈ 10,000 endotoxin units (EU)/mL of LPS from *E. coli* O111:B4 in water was used first. Therefore, anti-Lipid A particles were incubated with 10,000 EU/mL of LPS in water for 20 min at RT, followed by centrifugation. The particle pellet was resuspended with a fresh LPS solution, exposing particles to another round of 10,000 EU/mL of LPS in water. A total of 12 consecutive cycles were performed, as shown in the scheme in Figure 2a. The remaining amount of endotoxin in each SN was quantified using the traditional LAL test based on calibration curves (Appendix A), as well as with NTA (Figure 2b).

The amount of LPS detected in the SN after the first removal cycle was under the limit of detection for the LAL test (<0.1 EU/mL), which translates to a removal of at least 99.999%. From cycles 2 to 5, the removal was ≥99.99% according to LAL. In subsequent incubation cycles, the LPS removal performance of anti-Lipid A particles decreased continuously, indicating the stepwise saturation of the particle surface with LPS and their limit in effective further LPS removal (Figure 2b, blue curve).

Similarly to other amphiphilic molecules, LPS can form supramolecular structures in aqueous solutions [23,24,25]. The presence of pre-micelle LPS oligomers were already described in the literature by Santos et al. using light-scattering spectroscopy in LPS evaluation below the critical aggregation concentration [25]. To determine the sensitivity of NTA for LPS detection in the SN, a dilution series of LPS in water ranging from 1 to 1 × 10^6^ EU/mL was prepared (Appendix A). Unexpectedly, the results indicated that even at the lowest tested concentration (1 EU/mL ≈ 0.1 ng/mL), supramolecular structures were observed by NTA, providing enough sensitivity to determine the LPS removal efficiency of anti-Lipid A particles. 

The SN was then analyzed using NTA, measuring the number of particles/frame (Appendix A) and calculating the endotoxin removal for each cycle relative to the LPS starting concentration A (Figure 2b and Appendix A). While almost no LPS was detected in the SN during the first three cycles, a decrease in LPS removal performance was observed above 30,000 EU, followed by a continuous decrease during subsequent cycles (Figure 2b, black curve).

The curves obtained with LAL (enzymatic method) and NTA (physical method) are in close correspondence between cycles 1 and 8, demonstrating that both methods are well-suited for quantifying LPS removal from water. Starting at cycle 9, the results measured between the two methods started diverging. This can be explained considering that the NTA detection method is based on optical detection of supramolecular structures, while LAL is not limited to these aggregates. Nevertheless, NTA provided a faster and less expensive initial analysis of LPS solutions without the requirement of extensive sample dilution, as is common practice for LAL. Therefore, this technique can potentially be explored, complementary to LAL, in samples without light-scattering molecules.

Overall, our data showed a quantitative removal of more than 98% during the first three cycles evidenced by NTA and the first five cycles by LAL (Figure 2b and Appendix A). This indicates a saturation of 5 mg of particles with LPS between 30,000 and 50,000 EU; thus, at least 6000 to 10,000 EU can be removed with only 1 mg of anti-Lipid A particles (which corresponds to roughly 6.7 × 10^7^ particles). 

#### 2.2.2. Longitudinal Stability

In order to evaluate the stability of the Ab-particle conjugates, anti-Lipid A particles were stored at 2–8 °C and LPS removal performance was tested at specific time points between 8 and 251 days. As described for previous experiments, 5 mg of particles was incubated with a fresh solution of 10,000 EU/mL LPS in water at the indicated time points, and the corresponding SN was analyzed for traces of remaining LPS. Endotoxin levels for all SNs were below the limit of detection of the LAL assay, presenting less than 0.1 EU/mL even after more than 250 days of particle storage (Figure 2c). With that, we conclude that particles can be stored for at least 8 months at 2–8 °C (and potentially longer) without any measurable loss in functionality.

### 2.3. LPS Removal and Protein Recovery

Furthermore, the LPS removal performance of anti-Lipid A particles was compared with commercially available Endotoxin Removal Beads from Miltenyi Biotec [26]. Since these particles remove endotoxins through electrostatic interactions [6,26], in contrast to antibody–antigen interactions, they are here referred to as non-specific particles. Both particle types were tested in direct comparison in LPS solutions that additionally contained proteins ranging from low complexity (BSA in water) to medium (insulin in HEPES 25 mM pH 7.1) and high (BPE, naturally contaminated with unknown LPS strain in Dulbecco’s Phosphate-Buffered Saline, or DPBS). To evaluate differences in the performance of the two particle platforms, LPS removal efficiency was evaluated with LAL (BSA and insulin formulations) or with HEK-Blue™-hTLR4 cells (BPE formulation) and protein recovery was assessed with the BCA assay (BSA and insulin formulations) or Bradford assay (BPE formulation).

#### 2.3.1. LPS Removal and Protein Recovery in 1 mg/mL BSA in Water

As BSA is excipient for several formulations and pharmaceutical products [27,28] and, therefore, is a widely used model protein for the evaluation of new endotoxin removal methods [13,15,16], the LPS removal performance of anti-Lipid A particles and non-specific particles was first evaluated in the presence of BSA in water. 

Here, different stock solutions of LPS from *E. coli* O111:B4 (100 to 10,000 EU/mL) containing 1 mg/mL BSA were prepared. A total of 5 mg of either anti-Lipid A particles or non-specific particles were used to resuspend 1 mL of such LPS-BSA solutions. After incubation, particles were centrifuged, and the SNs were evaluated using LAL and BCA for LPS and protein content, respectively. Our results showed no significant difference in the LPS removal performance between both particle platforms, independent of the LPS content (Figure 3a). Nevertheless, the variability in the removal performance (assessed using the standard error of the mean, sem) was larger for the non-specific particles than for the anti-Lipid A particles, which indicates that the reproducibility of the LPS removal system based on electrostatic interactions was lower compared with the specific antibody–antigen interaction under these conditions. Furthermore, we noticed that LPS removal performance was lower for samples containing < 10,000 EU/mL. We rationalized that this is likely related to the fact that albumin has been described to strongly interact with LPS via hydrophobic interactions between the acyl chains of Lipid A with one or more hydrophobic binding sites in albumin [29]. Therefore, LPS molecules can be partially blocked by BSA, decreasing the interaction of LPS and the anti-Lipid A Ab, and, consequently, LPS is not properly removed. A similar result was recently described by Chen et al., who observed lower LPS removal as the ratio of human serum albumin to LPS increased [17]. Moreover, given the strong positive net charge of non-specific particles, it is also likely that BSA, which has a pI ≈ 4.7, adsorbs to the surface, forming a protein corona which hinders LPS binding and, thus, reduces LPS removal [14].

As previously mentioned, protein recovery is a crucial index to the selection of an endotoxin removal system, where the unwanted non-specific removal of protein components should be minimized or, ideally, avoided. When comparing the protein recovery performance of both particle types, a statistically significantly higher protein recovery yield was observed upon use of anti-Lipid A particles (Figure 3b). Notably, for these particles, the protein recovery slightly increased with higher LPS concentrations (93.2%, 96.8% and 97.8% for 100, 1000 and 10,000 EU/mL, respectively), while the non-specific particles did not follow this trend (84.0%, 88.1% and 87.3% for 100, 1000 and 10,000 EU/mL, respectively). The results obtained here highlighted the role of the specificity layer in reducing the non-specific removal of off-target molecules, allowing for higher protein recovery. 

#### 2.3.2. LPS Removal and Protein Recovery in 1 mg/mL Insulin in HEPES 25 mM

With the advent of recombinant DNA technology, insulin started to be produced in bacteria, more specifically in *E. coli*, and was the first FDA-approved recombinant drug [30,31]. To ensure safety for humans, the endotoxin level in insulin needs to be well-controlled, with the threshold of a specific insulin formulation being dependent on its potency and the recommended dose [7,32]. 

We therefore chose to investigate whether anti-Lipid A particles could be used for LPS removal in the presence of insulin in buffer. Similarly to the previous experiment performed with BSA in water, different amounts of LPS from *E. coli* O111:B4 were spiked with insulin (final insulin concentration was 1 mg/mL) and the SNs were evaluated for LPS and protein content. No significant difference was observed between the two different particle types regarding LPS removal (Figure 4a). Here, differently from previous experiments with BSA, the LPS contamination level did not influence the removal performance of particles, presumably due to lower non-specific interactions between insulin and LPS as compared with BSA-LPS. 

The BCA assay showed a higher statistically significant protein (insulin) recovery when using anti-Lipid A particles compared with non-specific particles in two out of three conditions (Figure 4b). In fact, anti-Lipid A particles achieved up to 100% protein recovery for 100 and 10,000 EU/mL, while non-specific particles achieved a maximum recovery of 92.9%, for 100 EU/mL. Although not significant, the mean protein recovery for 1000 EU/mL was still higher for the specific anti-Lipid A particles (99.6%) than for the non-specific particles (90.6%), consistent with the other two samples and the original intention and rationale of our specific anti-Lipid A particle platform. 

#### 2.3.3. LPS Removal and Protein Recovery in Naturally LPS-Contaminated BPE in DPBS

Given the intrinsic complexity of LPS-contaminated protein solutions, endotoxin removal systems are mainly tested in controlled samples where a known amount and serotype of LPS is spiked in a sample. For practical applications, there is an interest to test endotoxin removal from higher-complexity matrixes, such as BPEs, which represents a naturally LPS-contaminated sample. Birch pollen is the main cause of allergic rhinitis in northern and central Europe [33], containing more than 150 allergens including the protein Bet v 1 [33,34]. Upon extraction of birch pollen, BPE contains glycans as well as low-molecular-weight compounds, including LPS originating from different bacterial species encountered on the birch pollen surface [35,36]. To isolate the immune response from allergens and to help clarify their role, LPS should be removed from the extracts before exposing it to cells. As BPE is a genuine nature-derived solution, LPS contamination is of an unknown source, adding another level of complexity to our study. Initial LPS contaminations were measured to be ≈ 2.3 EU/mL. Thus, they were much lower than the spiked samples in previous experiments (the lowest tested condition was 100 EU/mL).

Here, we incubated 1 mL of BPE solution with 5 mg of either non-specific particles or anti-Lipid A particles. Due to interferences of BPE with the LAL assay, a cell-based assay was employed instead for the endotoxin quantification in the processed SN samples. The HEK-Blue™-hTLR4 cells were stimulated in the presence of LPS, triggering the inducible expression of secreted embryonic alkaline phosphatase (SEAP). SEAP was then determined using HEK-Blue™ detection. Endotoxin removal was measurable after incubation with both particle types, albeit in a lower range (ca. 20%), with no significant difference between the two systems (Figure 5a). Some environmental bacteria have the capability to produce LPS with uncommon chemical structures, including modifications in the Lipid A region [37]. Therefore, in addition to the fact that the BPE contains LPS from unknown bacterial sources, which might not be recognized by our anti-Lipid A antibody, further non-specific interactions between BPE constituents and LPS that would, similarly to BSA, hinder the LPS removal cannot be excluded.

However, when evaluating the protein recovery, the difference between using anti-Lipid A particles and non-specific particles was striking. Anti-Lipid A particles were able to recover up to 85% of the original proteins in the solution, whereas the non-specific particles only recovered around 50% (Figure 5b). A possible explanation for this observation could be the formation of a protein corona on the surface of charged non-specific particles, which would also reduce the interaction with and, consequently, the removal of LPS. 

## 3. Materials and Methods

### 3.1. Materials

Carboxy polystyrene particles of 3.00 µm (PC05003, Lot 10014) were acquired from Bangs Laboratories Inc. (Fishers, IN， USA). Phosphate-buffered saline (PBS), succinimidyl-4-(N-maleimidomethyl)cyclohexan-1-carboxylat (SMCC), aminodextran (AmDex), PBS-5 mM EDTA (PE) and Endotoxin Removal Beads (130-093-657, here also referred to as ‘non-specific particles’) were produced by Miltenyi Biotec. 

Endotoxin-free water (W50-1000) was purchased from Lonza. LPS from *E. coli* O111:B4 (L2630-25MG), insulin (I0516-5ML), N-Hydroxysuccinimide (NHS, 8045180025), tris(2-carboxyethyl)phosphine hydrochloride (TCEP, 75259-1G), N-Ethylmaleimide (NEM, 04259-5G), N-(3-Dimethylaminopropyl)-N′-ethylcarbodiimide hydrochloride (EDC, 03450-5G), sodium azide (SIAL71289-250G), dimethyl sulfoxide (DMSO, D2650-5X10mL), HEPES (H0887-20ML), Dulbecco’s Phosphate-Buffered Saline (DPBS, D8537-500ML) and Dulbecco’s Modified Eagle’s Medium—high glucose (DMEM, D5671-500ML) were acquired from Sigma-Aldrich. Endpoint chromogenic LAL test (LAL, A39553), polyclonal Goat anti-Lipid A antibody (anti-Lipid A, PA1-73178), Pierce™ BCA Protein Assay (23227), Pierce™ BCA Protein Assay Kit—Reducing Agent Compatible (23250), Coomassie Protein Assay Reagent (1856209) and *Betula pendula* (batch No 012519101) were purchased from Thermo Fisher Scientific. Donkey anti-goat-phycoerythrin antibody (anti-Goat-PE, 705-115-147) and goat anti-human-phycoerythrin (anti-Human-PE, 109-116-098) were purchased from Jackson Immuno Research. Bovine serum albumin (BSA, P6154-500GR) was purchased from BioWest. Pluronic F-68 (A1288,0100) was purchased from PanReac AppliChem. β -mercaptoethanol (β-ME, 444203) was acquired from Calbiochem. Copper grid (01801) was acquired from Ted-pella. Normocin (ant-nr-1), HEK-Blue™ Selection (hb-sel), HEK Blue™hTLR4 cells (hkb-htlr4) and HEK Blue™detection (hb-det2) were obtained from Invivogen. 

All plasticware used was sterile and either endotoxin-free certified or tested for the experiments. 

The following equipment was used during this study: Synergy H1 microplate reader from Biotek (Winooski, VT, USA) using software Gen5 v3.04; Countess II FL from Invitrogen (Waltham, MA, USA); NTA LM10 and its software NanoSight v3.2.16 from Malvern instruments Ltd. (Malvern, UK); MACSQuant 10 and its software MACSQuantify v2.13; and FEI Magellan 400L XHR SEM.

### 3.2. Methods

When not described differently, washing and centrifugation steps refer to 3000× *g*, 5 min, RT. An amount of 0.1 ng/mL of LPS from *E. coli* is considered to present a potency of 1 EU/mL [6]. Therefore, 1000 ng/mL LPS from *E. coli* O111:B4 is considered the same as 10,000 EU/mL in this study.

#### 3.2.1. Anti-Lipid A Particles’ Conjugation

The 10 mg carboxy PS particles were initially washed once with ethanol 70% to remove potential contamination and a second time with PBS-0.03% Pluronic F-68. Particles were then centrifuged, and the pellet resuspended in 0.5 mL EDC (20 mg/mL) and 0.5 mL NHS (12 mg/mL) and incubated for 15 min. The sample was then centrifuged, resuspended in an aqueous aminodextran solution (33 mg/mL) and incubated in an overhead mixer. After 2 h, particles were washed and resuspended in 1 mL PBS-0.03% Pluronic F-68 containing 30 µL of SMCC in DMSO and incubated for 1 h. Then, coupling of the anti-Lipid A antibody was performed via Michael addition, as previously described [38,39,40]. Here, the anti-Lipid A Ab was reduced for one hour with 1.5 mM TCEP prior to the addition to SMCC-activated particles. After the lapse of 2 h, the reaction was stopped via addition of 50 mM β-ME followed by 40 mM NEM. The particles were then washed 3 times and the first SN was saved for protein quantification. The final pellet was resuspended in PBS-0.03% Pluronic F-68 and sodium azide was added to a final concentration of 0.05%. Particles were stored at 2–8 °C until further use. 

#### 3.2.2. Anti-Lipid A Particles’ Characterization 

After the conjugation, the SN containing the unconjugated Ab was quantified via reducing-agent-compatible BCA, using anti-Lipid A Ab for the calibration curve. The amount of conjugated Ab was calculated by subtracting the total amount used for conjugation from the amount calculated with the BCA test.

Additionally, anti-Lipid A particles were stained with secondary antibodies for 20 min in the dark and washed with PE buffer to remove the unbound secondary Ab. As the anti-Lipid A Ab is a goat-raised Ab, the anti-Goat-PE should recognize its epitope and increase the MFI in the B2/PE channel (filter 585/40 nm) in MACSQuant 10. On the other hand, the anti-Human-PE should not bind to the anti-Lipid Ab conjugated on the particles and, therefore, not show any signal in the B2/PE channel. Flow cytometry data were analyzed in MACSQuantify. 

To evaluate the aggregation state of the conjugates, particles were optically imaged in solution in Countess Cell Counter (dilution 1:200 in PBS-0.03% Pluronic F-68). Moreover, in addition to determining the percentage of single particles in flow cytometry through FSC-A x FSC-H [41], we additionally gated the main population of single particles on FSC-A x SSC-A plots, with FSC/SSC set to log scale (Appendix A). 

Morphological characterization of microparticles was performed on an FEI Magellan 400L XHR SEM operating at 20 kV. Samples (10 μL) were deposited on an ultrathin formvar-coated 200-mesh copper grid. More than 100 particles were analyzed from STEM images with ImageJ to obtain the particle size distribution profile. 

#### 3.2.3. LPS Removal Capacity of Anti-Lipid A Particles

To assess the ability of anti-Lipid A particles to remove LPS, a saturation curve was generated. In short, a 1000 ng/mL ≈ 10,000 EU/mL solution of LPS from *E. coli* O111:B4 was prepared in endotoxin-free water via serial dilution of 5 mg/mL to 100 µg/mL, then to 1000 ng/mL. Between each dilution step, LPS was left 15 min in an overhead mixing at RT for homogenization and subsequently vortexed for 2 min. A total of 5 mg of anti-Lipid A particles were washed twice in water and then resuspended in 1 mL of LPS solution. The vial was incubated for 20 min using an overhead mixer. After 20 min, the sample was centrifuged, and the SN was saved. The pellet was resuspended again in 1 mL of LPS solution, and the procedure was repeated 12 times. For LPS quantification on the SN, the traditional endpoint chromogenic LAL test was employed following the manufacturer’s instructions, always using a freshly prepared calibration curve (Appendix A) and diluting samples as necessary. All SNs were previously filtered with a 0.65 µm sterile centrifugal filter to remove leftover particles and prevent interferences with the LAL. After that, the amount of LPS calculated in the SN (Appendix A) was compared with the initially added (10,000 EU/mL) and the endotoxin removal percentage was plotted (Equations (S1) and (S2)). Complementarily, due to the amphiphilic nature of LPS and its tendency to form supramolecular structures, these SNs were also characterized using NTA. The NanoSight software automatically reports the number of particles/frame for each sample. It is important to note that the settings for processing (screen gain and threshold) were adjusted for the control samples and kept constant for further measurements. The LPS removal percentage was calculated considering the number of particles/frame in each SN and the number of particles/frame in the reference solution (Appendix A and Equations (S2) and (S3)). Here, 0% removal is associated with SN with the number of particles/frame equal to the number of particles/frame in the 10,000 EU/mL control. 

#### 3.2.4. Longitudinal Stability

To evaluate the Ab-conjugated particles’ stability over time, anti-Lipid A particles were stored at 2–8 °C at a concentration of 20 mg/mL in PBS-Pluronic 0.03% with 0.05% sodium azide. At different time points (i.e., days 8, 14, 25, 34, 40, 41, 60, 77, 156, 161 and 251), particles were centrifuged and incubated with LPS from *E. coli* O111:B4 in water. In short, a solution of 1000 ng/mL of LPS was prepared as previously described. Then, 1 mL of the LPS solution was added to 5 mg of anti-Lipid A particles and incubated in an overhead mixer for 20 min at RT. The sample was then centrifuged, and the SN was collected and filtered with a 0.65 µm centrifugal filter before endotoxin quantification using LAL test.

#### 3.2.5. LPS Removal and Protein Recovery in 1 mg/mL BSA in Water 

Different concentrations of LPS from *E. coli* O111:B4 were prepared (1000 ng/mL ≈ 10,000 EU/mL; 100 ng/mL ≈ 1000 EU/mL; 10 ng/mL ≈ 100 EU/mL). Dried BSA was weighted and used without previous LPS detoxification and resuspended in either LPS-free water or directly using the different LPS solutions to achieve a consistent final BSA concentration of 1 mg/mL in all cases. The intrinsic LPS content in 1 mg/mL BSA in pure water was determined to be 14 EU/mL. All samples were mixed in an overhead mixer for at least 15 min at RT before usage. A total of 5 mg of anti-Lipid A particles and 5 mg of non-specific particles were washed twice with water and finally resuspended in 1 mL of the BSA-LPS solutions. Vials were incubated for 20 min in an overhead mixer at RT. After 20 min, samples were centrifuged, and the SNs were filtered through a 0.65 µm sterile centrifugal filter. After that, each SN was subjected to LAL and BCA assays to evaluate its endotoxin content and protein concentration, respectively. For the LAL, samples were diluted as necessary to ensure a concentration < 50 µg/mL of BSA in order to avoid interferences [42]. The LPS removal performance was calculated considering the initial endotoxin contamination of the sample (10,000 EU/mL, 1000 EU/mL or 100 EU/mL) and the resulting concentration determined in the acquired SN via the LAL test. The protein recovery was calculated comparing the initial protein content to the final concentration after the removal process, both assessed via BCA. 

#### 3.2.6. LPS Removal and Protein Recovery in 1 mg/mL Insulin Solution in HEPES 25 mM

Insulin stock solution was used without detoxification, with an intrinsic LPS concentration of less than 0.1 EU/mL at a final concentration of 1 mg/mL in HEPES buffer 25 mM, pH 7.1. Insulin was diluted with LPS from *E. coli* O111:B4 to achieve a final concentration of 1 mg/mL insulin containing 1000 ng/mL ≈ 10,000 EU/mL, 100 ng/mL ≈ 1000 EU/mL or 10 ng/mL ≈ 100 EU/mL in HEPES 25 mM, pH 7.1. Solutions were kept in an overhead mixer for at least 15 min at RT before usage. A total of 5 mg of anti-Lipid A particles and 5 mg of non-specific particles were washed twice with HEPES 25 mM pH 7.1 and resuspended in 1 mL of the insulin-LPS solutions. Vials were incubated for 20 min in an overhead mixer at RT. After 20 min, all samples were centrifuged, and SNs were further filtered with a 0.65 µm sterile centrifugal filter. After that, SN were evaluated regarding endotoxin and protein content as described in Section 3.2.5.

#### 3.2.7. LPS Removal and Protein Recovery in Naturally LPS-Contaminated BPE in DPBS 

Birch pollen extract was prepared based on the protocol in Johnson et al. with slight modifications [43]. In total, 100 mg of *Betula pendula* pollen was dispersed in 1 mL of DPBS. This dispersion was incubated at 4 °C in an overhead mixer for 18 h to allow for the extraction of a plethora of substances, including LPS. After that, the vial was centrifuged at 4 °C, 14,000 rpm for 30 min. The SN was filtered with a 0.22 µm filter and frozen at −20 °C until use. 

For LPS removal, a 1:10 dilution of the above-mentioned BPE solution was prepared in DPBS. Briefly, 5 mg of anti-Lipid A particles and 5 mg of non-specific particles were washed twice with DPBS and resuspended with 1 mL of BPE. The vial was incubated for 20 min in an overhead mixer at RT. After 20 min, samples were centrifuged, and the SN was further filtered with a 0.65 µm sterile centrifugal filter to remove potential remaining particles. Due to interferences of BPE with the LAL assay, a cell-based assay was employed for the endotoxin quantification in the obtained SN. HEK-Blue™-hTLR4 cells were used for endotoxin quantification in BPE, based on a protocol provided by Johnson et al. [43] with slight modifications. The cells were cultured in DMEM supplemented with penicillin-streptomycin, glutamine, normocin, inactivated FBS and HEK-Blue™ Selection. Upon stimulation of cells with LPS, an inducible expression of SEAP was triggered and determined using HEK-Blue™ detection. In short, 20 µL of blank (water), negative control (DPBS), positive control (10 EU/mL), standards (calibration curve), sample control (BPE) and the SN after the removal process were added to a 96-well plate and incubated for 10 min at 37 °C, 5% CO_2_. When that time elapsed, 180 µL of 2.5 × 10^5^ cells/mL in supplemented DMEM without HEK-Blue™ Selection were added to each well. The plate was then incubated for 22 h at 37 °C, 5% CO_2_. On the following day, 20 µL of the SN were transferred to a clean 96-well plate and 180 µL of HEK Blue™ detection was added to each well. The plate was further incubated for 6 h at 37 °C, 5% CO_2_. After that time, the absorbance at 650 nm was measured in a plate reader. The endotoxin quantified on samples was compared with the initial endotoxin content and the removal percentage was calculated (Equations (S2) and (S4)). In a similar manner, BPE interferes in the BCA assay for protein quantification; therefore, the Bradford assay was applied for this purpose. Here, 40 µL of blank (water), negative control (DPBS), standards (calibration curve), sample control (BPE) and the SN after the removal process were added to a 96-well plate and 160 µL of Coomassie was added to each well. The absorbance at 595 nm was measured and the protein concentrations were calculated based on a BSA calibration curve. The protein recovery was calculated comparing the initial protein content to the final concentration after the removal process, both assessed via Bradford assay. 

## 4. Conclusions

Endotoxin removal from sensible samples is a challenging process due to the wide range of possible interactions that these amphiphilic molecules can have with sample constituents. Non-specific endotoxin removal methods can be quite effective, resulting in samples with acceptable endotoxin contents. However, the loss of non-target active components can be a limitation for some applications. 

In this work, we showed how the conjugation of polystyrene microparticles to a polyclonal anti-Lipid A antibody can help to avoid the non-specific depletion of other active molecules while achieving a similarly high LPS removal in comparison with non-specific removal particles. This was tested by measuring the endotoxin removal and protein recovery performance over a series of solutions with increasing matrix complexity from water < BSA in water < insulin in HEPES 25 mM < BPE in DPBS, where the latter was a naturally LPS-contaminated sample. In addition to the well-established LAL assay, we, to the best of our knowledge, were the first to use NTA as an alternative method to characterize LPS, with results corresponding to those obtained with the LAL assay. It should be noted that in our study NTA was suitable for the detection of LPS supramolecular structures only if no other light-scattering molecules were present (i.e., LPS formulations in pure water). The potential use of NTA in complex LPS samples is still to be explored and developed. Overall, we observed that, independent of the buffer and sample constituents, anti-Lipid A particles enhanced the protein recovery for all conditions compared with non-specific particles. In fact, the advantage of the antibody–antigen interactions was most prominent in samples containing BPE, where the protein recovery was 35% higher for anti-Lipid A particles. On the other hand, while one would assume that high LPS concentrations are the weak point of particle-based removal systems given their surface saturation, we observed that LPS removal was challenging at LPS concentrations ≤ 100 EU/mL, where LPS interactions and competition on the binding sites by other molecules play a crucial role [14]. Overall, we present a new system that allows for efficient and specific LPS removal within 20 min, while maintaining protein recoveries > 85%, even in complex solutions such as BPE.

## Figures and Tables

**Figure 1 ijms-24-13971-f001:**
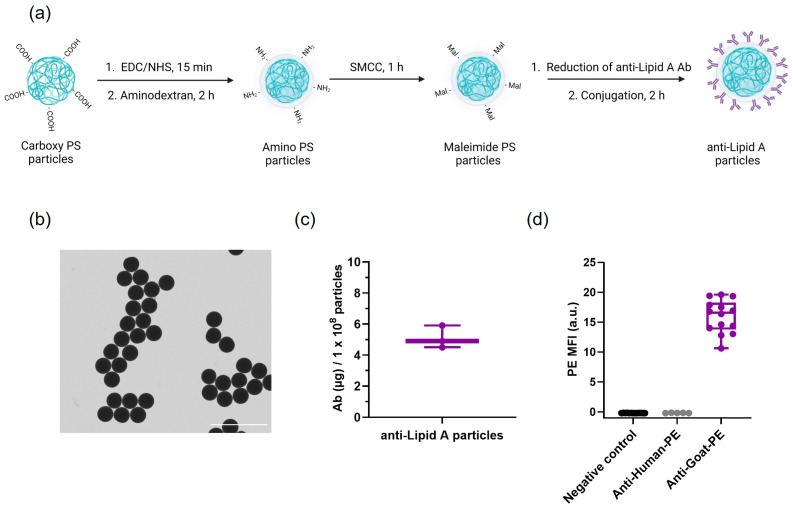
Anti-Lipid A particles: Ab conjugation and characterization. (**a**) Schematic representation of the anti-Lipid A particles’ conjugation workflow. Anti-Lipid A particle characterization: (**b**) STEM image at 8000× magnification (scale bar: 10 µm); (**c**) quantification of anti-Lipid A antibody loading normalized to 1 × 10^8^ particles (n = 3); and (**d**) flow cytometry median fluorescence intensity (MFI) of PE-secondary Ab staining of anti-Lipid A Ab on particles.

**Figure 2 ijms-24-13971-f002:**
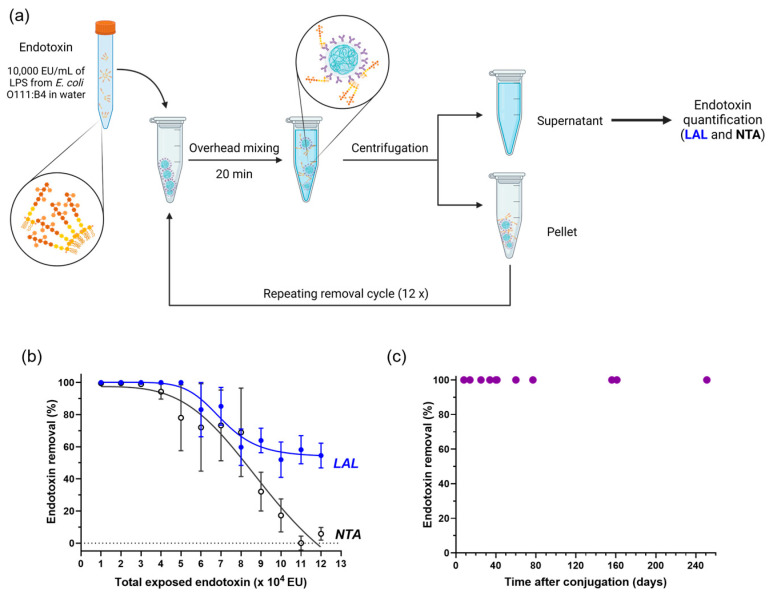
LPS removal capacity and longitudinal stability of anti-Lipid A particles. (**a**) Schematic representation of the LPS removal procedure based on repeating removal cycles, and the analysis of SN using LAL and NTA. (**b**) LPS saturation curve obtained by exposing 5 mg of anti-Lipid A particles to repeated cycles of 10,000 EU of LPS from *E. coli* O111:B4 in water (mean ± sem, n ≥ 3). (**c**) LPS removal performance of anti-Lipid A particles when incubated with 10,000 EU/mL of LPS from *E. coli* O111:B4 measured over a period of 240 days and stored at 4 °C.

**Figure 3 ijms-24-13971-f003:**
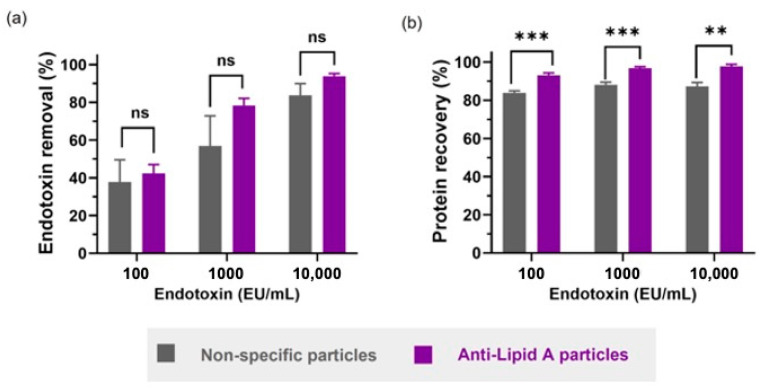
Comparison between anti-Lipid A particles and commercial non-specific particles in aqueous BSA solutions regarding (**a**) endotoxin removal and (**b**) protein recovery. All samples contain 1 mg/mL of BSA in water with different spiked amounts of LPS from *E. coli* O111:B4. Data shown are mean ± sem, n ≥ 3, and analyzed via *t*-test (ns = not significant, ** *p* < 0.01, *** *p* < 0.001).

**Figure 4 ijms-24-13971-f004:**
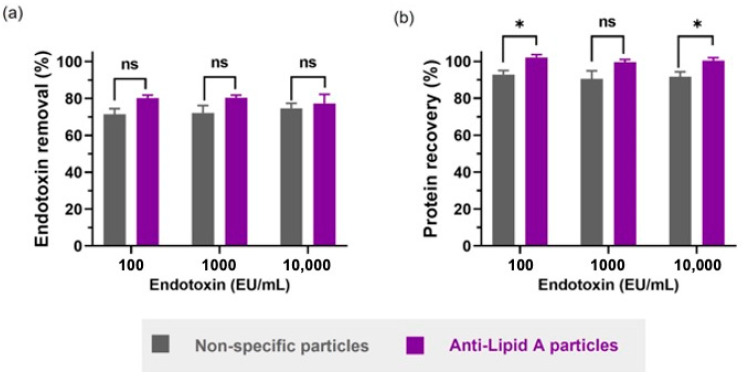
Comparison between anti-Lipid A particles and commercial non-specific particles in insulin in HEPES buffer regarding (**a**) endotoxin removal and (**b**) protein recovery. All samples contain 1 mg/mL of insulin in HEPES 25 mM pH 7.1 with different spiked amounts of LPS from *E. coli* O111:B4. Data shown are mean ± sem, n ≥ 3, and analyzed with *t*-test (ns = not significant, * *p* < 0.05).

**Figure 5 ijms-24-13971-f005:**
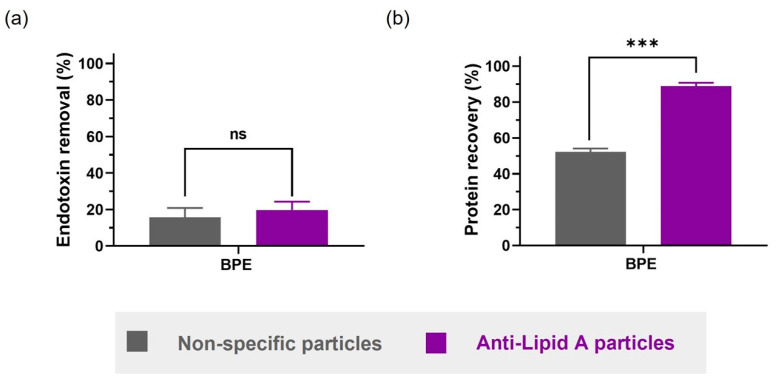
Comparison between anti-Lipid A particles and commercial non-specific particles in BPE in DPBS regarding (**a**) endotoxin removal and (**b**) protein recovery. Samples are solutions of BPE in DPBS with natural LPS contamination of unknown source. Data shown are mean ± sem, n ≥ 3, and analyzed with *t*-test (ns = not significant, *** *p* < 0.001).

## Data Availability

The data presented in this study are available on request from the corresponding author.

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
