# Peer review of "Recovering What Matters: High Protein Recovery after Endotoxin Removal from LPS-Contaminated Formulations Using Novel Anti-Lipid A Antibody Microparticle Conjugates"

_ijms, 2023, doi:10.3390/ijms241813971_

Round 1
Reviewer 1 Report
The manuscript “Recovering what matters: high protein recovery after endotoxin removal from LPS-contaminated formulations using novel anti-Lipid A antibody microparticle conjugates” by Cristiane Casonato Melo et al. describes the development of a novel method for selective removal of liposaccharides (LPS) from pharmaceutical formulations containing active proteins.
LPS or endotoxins, major components of the outer membrane of Gram-negative bacterial cell walls, play a key role in microbial pathogenesis. They are able to activate the human innate immune system, with symptoms ranging from fever to a septic shock or even death. In order to avoid contamination incidents, strict regulatory limits for LPS content apply to all injectable pharmaceuticals. The need for specific endotoxin removal methods became even more evident with the development of recombinant drugs, produced in a Gram-negative bacterium E. coli.
The majority of endotoxin removal systems, available to date, are based on non-specific interactions of LPS to positively charged or hydrophobic surfaces. However, the lack of specificity can result in unwanted losses of essential proteins from the pharmaceutical product.
The novel method, presented in the manuscript, is based on the specific antigen-antibody interactions between the lipid A moiety from LPS and an anti-Lipid A antibody immobilized on the surface of polystyrene microparticles. Lipid A is considered as a most conserved region of LPS among different bacterial species, which makes this method useful for a wide range of LPS contaminants.
Micron-size polystyrene (PS) particles were surface modified with commercially available anti-Lipid A antibodies via 3-step conjugation. The anti-Lipid A particles were imaged using scanning transmission electron microscopy (STEM), scanning electron microscopy (SEM) and cell counter; and analyzed by flow cytometry after staining with a secondary antibody carrying a fluorescent tag. The conjugated particles were shown as spherical with a narrow-size distribution, and highly dispersible with no visual signs of aggregation.
The LPS removal capacity and the saturation level of the conjugated particles was accessed by 12 consecutive cycles of incubation with E. coli O11:B4 LPS (10,000 ng/mL~10,000 EU/mL) in water. LPS remaining in the supernatants (SN) was quantified using two methods: a traditional Limulus Amebocyte Lysate (LAL) test and an alternatively developed method based on nanoparticle tracking analysis (NTA), detecting the supramolecular structures (micelles) of LPS. The latter method have been never applied to the quantification of LPS. The authors showed that even at the lowest tested LPS concentration (1 EU/mL≈ 0.1 ng/mL), supramolecular structures were observed by NTA, providing enough sensitivity to determine LPS removal efficiency of anti-Lipid A particles. NTA thus provided a faster and less expensive analysis of LPS solutions, compared to the LAL test.
Longitudinal stability studies of the anti-Lipid A particles showed that they could be stored for at least 8 months in the refrigerator without any measurable loss in functionality.
As a next step of method development, the anti-Lipid A particles were compared to commercially available “non-specific particles”—Endotoxin removal beads from Miltenyi Biotec. LPS removal capacity and protein recovery of two types of particles were assessed. Both particle types were tested in LPS solutions (100 to 10, 000 EU/mL) that additionally contained proteins with increased complexity: BSA (1 mg/mL) in water and insulin (1mg/mL) in HEPES buffer. Finally, the efficiency of both types of particles was compared in naturally LPS-contaminated Birch pollen extract (BPE) in DPBS buffer.
The study demonstrated that the two types of particles showed a similar endotoxin removal capacity. However, the anti-Lipid A particles showed a significantly higher protein recovery, especially in samples containing BPE.
The main findings of the current study consist in (i) creation of novel microparticles with conjugated anti-Lipid A antibodies, which can be successfully used for removal of LPS contaminations from various protein preparations, and (ii) application of NTA for the evaluation of LPS supramolecular structures and quantitative LPS measurements.
Despite the technical complexity of the subject, the article is clearly written and pleasant to read. The illustration schemes (Figs. 1a and 2a) are very helpful for a better understanding of the experimental methodology.
My comments are only minor.
1. Abstract. The authors may add additional information concerning the characterization of the microparticles.
2. Results and discussion. In Section 2.2., twelve consecutive removing cycles are described. The remaining amount of LPS in SN was assessed by two techniques, LAL and NTA. However it is not that clear what methodology was adapted for further studies, described in Section 2.3. I would suggest to add this information ex.at the end of the Section 2.3
Author Response
Thank you for your review notes and comments.
To reply on your minor changes:
1) We have implemented one sentence in the abstract to described the analytical techniques used.
2) regarding the clarity of techniques used from section 2.2 to section 2.3, we have added clarifying statements on lines 225 to 229.
We hope you find this suffice to address your comments to help make our manuscript more clear and complete.
Thank you for your comments!
Reviewer 2 Report
Very satisfactory article with multiple experiments concerning the construction of the beads for specific, anti-Lipid A antibody based removal of endotoxin from protein solutions with the maximal recovery of protein of interest.
Potentially interesting findings, concerning the efficacy of the constructed system and its storage stability, proven by convincing experiments. Also the impact of the presented results can be high, with potential new product with wide applicability.
Major remarks:
In my opinion NTA method will not be suitable for endotoxin content analysis in complex matrices. It was tested on pure LPS solutions, in the presence of other lipids, detergents or complex particles it will be not applicable. Moreover, in concentrations below CMC LPS should not form the micelles. Please discuss the findings described in paragraph in lines 165-170
It would be interesting to compare the constructed system with existing products with the specificity higher that ion-exchange beads used, like polymyxin immobilized resins. Please comment.
Was BSA preparation detoxified (cleared from endotoxins before experiments), or the endotoxin content measured in the stock BSA and insulin preparations (before LPS spiking)? It could influence the results of entoxin removal experiments.
Minor remarks:
Procedural remark: I am not sure, what is the influence of the described procedure in which dry BSA preparation is solubilized directly in target LPS solution, and where the BSA molecules are hydrated in the presence of relatively high LPS concentrations. I would rather mix two water solutions of that molecules, please comment that element of the procedure.
Was BSA preparation detoxified, or the endotoxin content measured in the stock BSA and insulin preparations (before LPS spiking)?
Line 140 - a typo: "Figure 2d" instead of "Figure 1d"
Line 202 - the bead source should be aldo mentioned in "Materials and Methods" section
Author Response
Thank you for your review notes and comments.
To reply on your major changes:
Thank you for pointing out the potential limitations of NTA. Indeed, NTA is sensitive to light-scattering molecules and this is critical when working with complex matrices. In our work, we have not explored the use of NTA in complex formulations. Thus, we have added additional clarifying statements in lines 360-363 and 193-195.
Regarding the CMC, we agree that the CMC defines the lowest concentration of micelles. However, it is known from other LPS molecules, that pre-micellar supramolecular structures can be formed below the CMC values. The presence of pre-micelles structures were already described in literature by Santos et al. 2003, using light scattering spectroscopy in LPS evaluation, therefore the supramolecular structures present in low concentrations are probably pre-micelle LPS oligomers. See Ref. 25: "Santos, N.C.; Silva, A.C.; Castanho, M.A.R.B.; Martins-Silva, J.; Saldanha, C. Evaluation of Lipopolysaccharide Aggregation by Light Scattering Spectroscopy. Chembiochem 2003, 4, 96–100." We therefore added a clarifying statement in lines 173-176 to our current manuscript.
Regarding, the comparison to polymyxin immobilized resins is definitely interesting and could, e.g. potentially be further explored with anti-Lipid A particles for hemoperfusion in sepsis treatment. Unfortunately, the comparison to polymyxin resin is out of scope in this work.
Regarding the comments on BSA: BSA or Insulin were not decontaminated before used. However, the intrinsic LPS content was determined in the stock solutions. This information is now added to the main manuscript in lines 485-489 and 507-509. The intrinsic amount of LPS found is low enough that we consider this should not influence the described results or interpretation.
Regarding the BSA mixing procedure, we have added additional statements in the M&M section of the manuscript, in hope to enhance the clarity of the text. We agree that is it definitely important to be consistent in used procedures. In our case, this is common practice in our labs and we have consistently followed the same procedure across all our studies. Please see lines 485-489.
Last, regarding the bead source, please see line 379.
We hope you find this suffice to address your comments to help make our manuscript more clear and complete.
Thank you for all your comments!
Reviewer 3 Report
Figure 2 = Could authors mention whether they mix/rotate samples during incubation of anti-lipid A particles and endotoxin?
Typo in the manuscript title of supplementary information.
Author Response
Dear Reviewer,
Thank you for your notes.
In reply to your comment regarding Figure 2, we have now updated this figure and include "Overhead mixing, 20 min." over the first arrow in this Figure.
Thanks for noticing the typo in the SI title.
We hope you find this suffice to address your comments to help make our manuscript more clear and complete.
Thank you for your comments!